# Review of Wafer Surface Defect Detection Methods

**Jianhong Ma, Tao Zhang, Cong Yang, Yangjie Cao, Lipeng Xie \*, Hui Tian \* and Xuexiang Li \***

School of Cyber Science and Engineering, Zhengzhou University, Zhengzhou 450000, China
\* Correspondence: lipengxie@zzu.edu.cn (L.X.); tianhui@zzu.edu.cn (H.T.); lxx@zzu.edu.cn (X.L.)

**Abstract:** Wafer surface defect detection plays an important role in controlling product quality in semiconductor manufacturing, which has become a research hotspot in computer vision. However, the induction and summary of wafer defect detection methods in the existing review literature are not thorough enough and lack an objective analysis and evaluation of the advantages and disadvantages of various techniques, which is not conducive to the development of this research field. This paper systematically analyzes the research progress of domestic and foreign scholars in the field of wafer surface defect detection in recent years. Firstly, we introduce the classification of wafer surface defect patterns and their causes. According to the different methods of feature extraction, the current mainstream methods are divided into three categories: the methods based on image signal processing, the methods based on machine learning, and the methods based on deep learning. Moreover, the core ideas of representative algorithms are briefly introduced. Then, the innovations of each method are compared and analyzed, and their limitations are discussed. Finally, we summarize the problems and challenges in the current wafer surface defect detection task, the future research trends in this field, and the new research ideas.

**Keywords:** wafer defect pattern; image processing; defect identification method; feature extraction

## 1. Introduction

Silicon wafers are used in the manufacturing of semiconductor chips. The required patterns are formed on the wafers through lithography and other processes and are very important carriers in the semiconductor chip manufacturing process. In the manufacturing process, due to the influence of factors such as the environment and process parameters, defects will be generated on the surface of the wafer, which will affect the yield of wafer production. The accurate detection of wafer surface defects can accelerate the identification of abnormal faults in the manufacturing process as well as the adjustment of the manufacturing process, improve production efficiency, and reduce scrap rates.

Early detection of wafer surface defects is often performed manually by experienced inspectors, which has problems such as low efficiency, poor accuracy, high cost, and strong subjectivity, which are insufficient to meet the requirements of modern industrialized products. At present, defect detection methods based on machine vision [1] have replaced manual inspection in the field of wafer inspection. Traditional machine vision-based defect detection methods often use manual feature extraction, which is inefficient. The emergence of computer vision-based detection methods [2], especially the advent of neural networks such as convolutional neural networks, has addressed the limitations of data preprocessing, feature representation and extraction, and model learning strategies [3]. With their high efficiency, accuracy, low cost, and strong objectivity, neural networks have rapidly developed and have been widely applied in the field of surface defect detection in semiconductor wafers.

In recent years, with the development of electronic integrated circuits, such as smart terminals and wireless communication facilities, and the promotion of Moore's Law [4], while the global demand for chips has increased, the accuracy of the lithography process has increased. With the advancement of technology, process precision has reached below

10 nm [5]. As a result, higher requirements have been placed on the yield of each process step, posing greater challenges for defect detection techniques in wafer manufacturing.

This paper mainly summarizes the related research on wafer surface defect detection algorithms, including traditional image processing, machine learning, and deep learning. According to the characteristics of the algorithm, the relevant literature is summarized and organized, and the problems and challenges faced in the field of wafer defect detection and future development are prospected. This paper intends to help in the quick understand of relevant methods and skills in the field of wafer surface defect detection.

## 2. Wafer Surface Defect Patterns

In actual production, there are many kinds of defects on wafers, and the shapes are not uniform, which increases the difficulty of wafer defect detection. Among the types of wafer defects, unpatterned wafer defects and patterned wafer defects are the two main forms of wafer defects. These two defects are the main cause of chip failure. Unpatterned wafer defects mostly occur in the pre-lithography stage of wafer production [6], which are wafer defects caused by machine failures. The scratch defect is shown in Figure 1a, and the particle contamination defect is shown in Figure 1b. Patterned wafer defects are mostly found in the middle process of wafer production. Improper exposure time, development time, and post-baking time lead to defects in photolithographic lines. Defects on the wafer surface generated during micro/nano-fabrication of spiral excitation coils and fork-shaped electrodes are shown in Figure 2. The open circuit defect is shown in Figure 2a, the short circuit defect is shown in Figure 2b, the line contamination defect is shown in Figure 2c, and the bite defect is shown in Figure 2d.

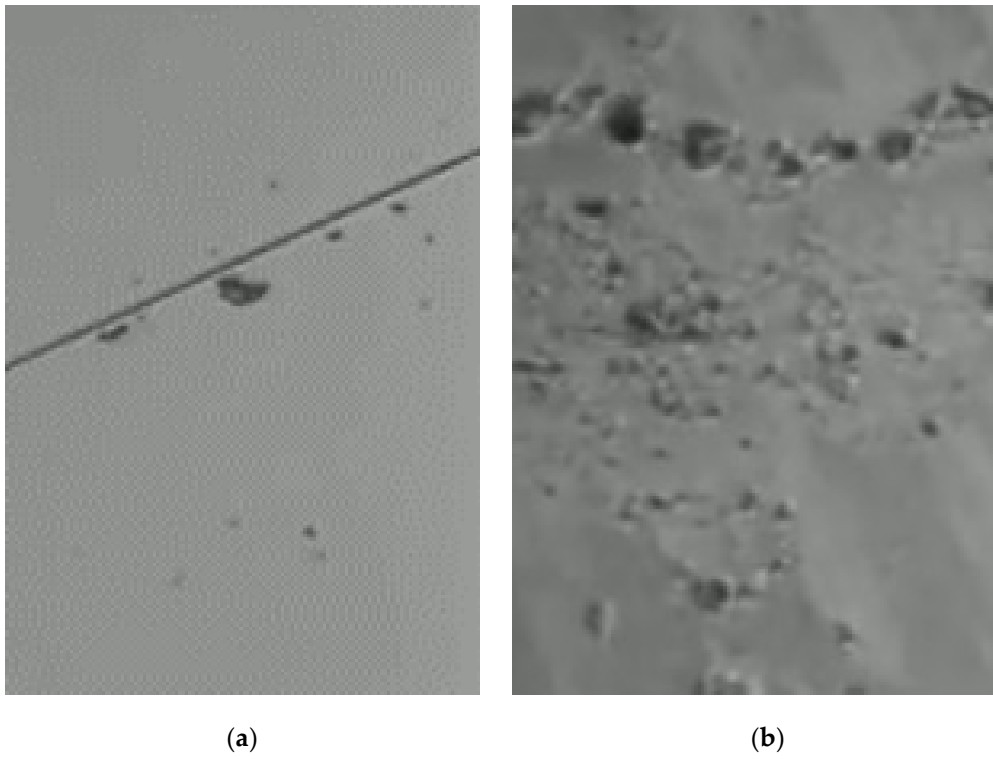

(**a**)  (**b**)

**Figure 1.** (**a**) Scratch defects in unpatterned wafers; (**b**) particulate contamination in unpatterned wafers.

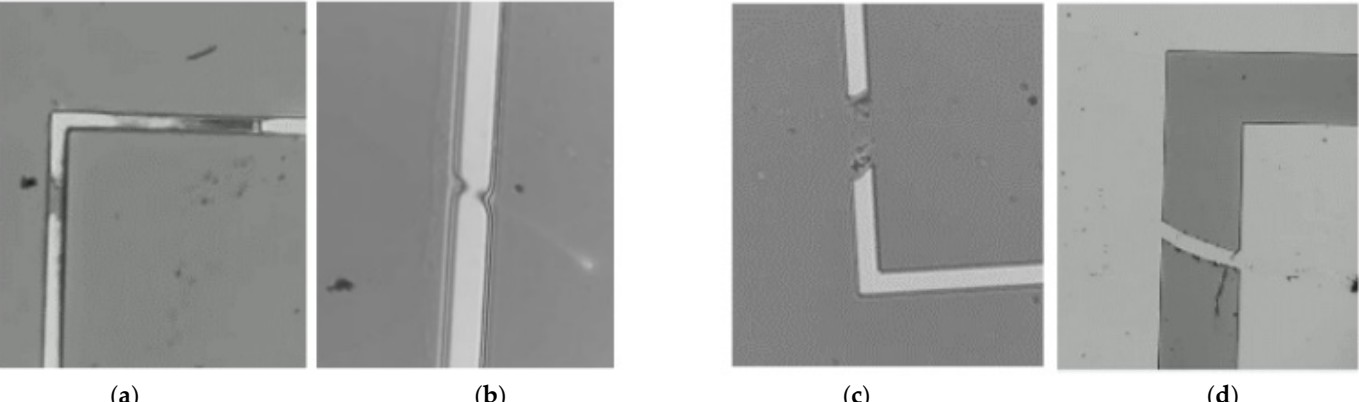

**Figure 2.** (**a**) Open defects, (**b**) short defects, (**c**) line contamination, and (**d**) bite defects in a patterned wafer defect map.

Due to the existence of the above-mentioned wafer defects, when the functional integrity test of all the chips on the wafer is performed, chip failures may occur. The chip engineer marks the test results with different colors to distinguish the position of the chip. Under the influence of different operating processes, corresponding specific spatial patterns will be generated on the wafer [7]. The wafer image data, i.e., the wafer map, is thus generated. As stated by Hansen et al. [8] in 1997, defective chips usually have a clustering phenomenon or show some systematic patterns, and this defect pattern usually contains the necessary information on process conditions [9]. The wafer map can not only reflect the integrity of the chip but also accurately describe the spatial position information corresponding to the defect data. Wafer maps may exhibit spatial dependencies across the wafer, and chip engineers can often trace the cause of defects and resolve problems based on defect type. Mirza et al. [10] divided wafer map defect patterns into general and local types, namely global random defects and local defects. The wafer map defect pattern diagram is shown in Figure 3, the local defects are shown in Figure 3a, and the global random defects are shown in Figure 3b. Global random defects are generated by uncertain factors, which are uncontrollable factors without specific clustering phenomena, such as dust particles in the environment. Global random defects can only be reduced through long-term, incremental improvements or expensive equipment overhaul programs. Localized defects are inherent to a system and are influenced by controllable factors during the wafer production process, such as process parameters, equipment issues, and improper operation. They appear on the wafer repeatedly and exhibit a certain level of clustering. Identifying and classifying local defects to locate equipment anomalies and inappropriate process parameters plays a vital role in improving the yield rate of wafer production.

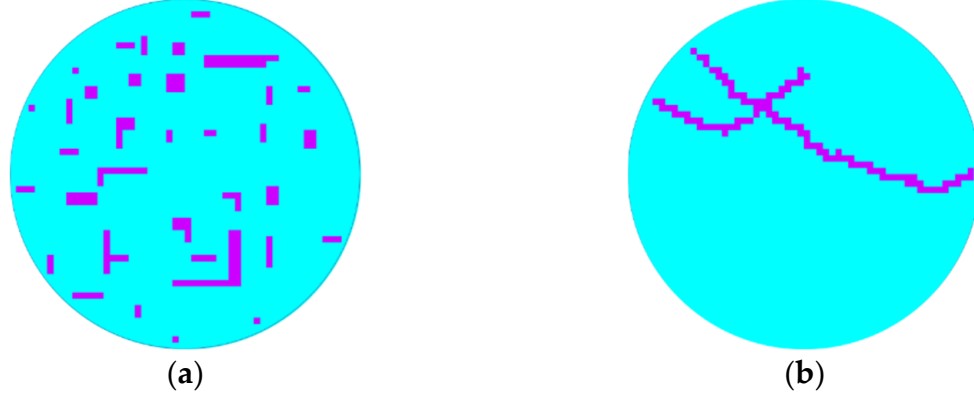

**Figure 3.** (**a**) Local defect pattern; (**b**) global defect pattern.



For wafer patterns with a large area, low feature size, low density, and low integration, the lithography path can be observed with an electron microscope, and trace detection can be performed directly. With the significant increase in the integration level of chip circuits, it has become increasingly difficult to perform chip-level inspection. This is because as integration level increases, the components on the chip become smaller, more complex, and more densely packed, resulting in a greater number of potential defects [11]. These defects are difficult to detect and repair through conventional inspection methods, requiring more sophisticated and advanced detection technologies and tools.

The research on wafer maps is a hot topic in wafer defect detection [12–15]. Liu Fengzhen of Tianjin University [16] researched wafer map defects caused by abnormal lithography equipment. For the defects in the actual production process of wafers, we have conducted in-depth research on photoresist, wafer dust particles, and wafer ring, scratch, spherical, linear, and other defects through equipment experiments, aiming to find the cause of defects to improve productivity. To determine the cause of wafer mode failure, Ming-Ju Wu [11] et al. collected 811,457 real wafer maps from actual manufacturing and created the WM-811K wafer map dataset, which is currently the most widely used wafer map. Semiconductor domain experts annotated eight defect pattern types for approximately 20 percent of the wafer maps in this dataset. The eight types of wafer map defect patterns are shown in Figure 4. Most of the articles cited in this review conducted their testing based on this dataset.

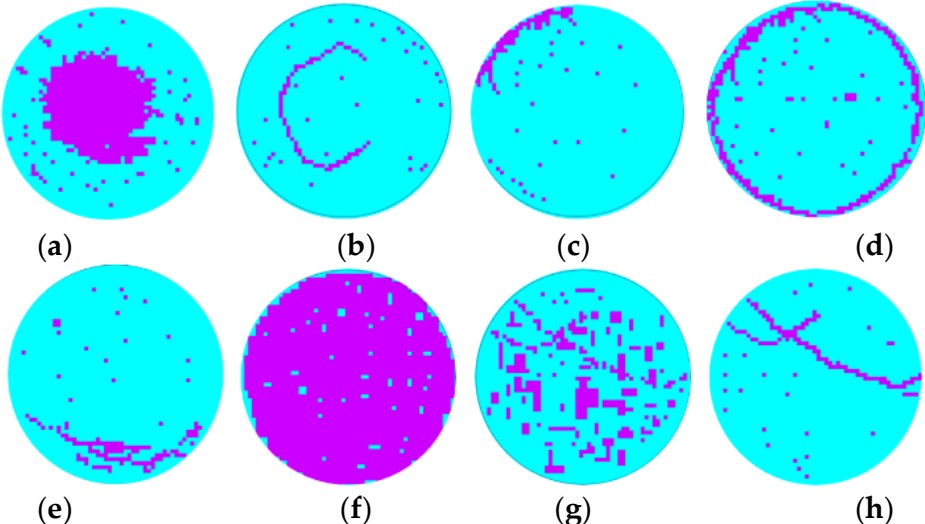

**Figure 4.** Eight types of wafer map defect pattern types: (**a**) center, (**b**) donut, (**c**) edge-loc, (**d**) edge-ring, (**e**) local, (**f**) near-full, (**g**) random, and (**h**) scratch.

## 3. Wafer Surface Defect Detection Based on Image Signal Processing

Image signal processing is used to convert image signals into digital signals, which are then processed via computer technology to achieve image transformation, enhancement, and detection [17–19]. Commonly used in the field of wafer inspection are wavelet transform (WT), spatial filtering (spatial filtering), and template matching (template matching). This section mainly introduces the application of these three algorithms on wafer surface defect detection. The comparison of image processing algorithms is shown in Table 1.

**Table 1.** Comparison of image processing algorithms.

| Model Algorithm | Innovation | Limitation |
| --- | --- | --- |
| Wavelet transform [12,20–24] | The image can be decomposed into multiple resolutions and presented as local sub-images with different spatial frequencies. Anti-grain. | The selection of the threshold is very dependent and the adaptability is poor. |

**Table 1.** *Cont.*

| Model Algorithm | Innovation | Limitation |
| --- | --- | --- |
| Spatial filtering [25–33] | Based on spatial convolution, remove high-frequency noise, and perform edge enhancement. | Performance depends on the threshold parameter. |
| Template matching [11,17,34–36] | The template matching algorithm has strong anti-noise ability and fast calculation speed. | Sensitive to feature object size. |

*3.1. Wavelet Transform*

Wavelet transform (WT) [20] is a signal time–frequency analysis and processing technology. Firstly, the image signal is decomposed into different frequency subbands through a filter to perform wavelet decomposition. Then, by calculating the mean, standard deviation, or other statistical measures of the wavelet coefficients, each coefficient is analyzed to detect any anomalies or defects. Anomalies or defects may manifest as sudden changes or outliers in the wavelet coefficients. Based on the analysis results, pre-defined thresholds are used to determine the defects and anomalies in the signal, and the location of the defect is determined by identifying which time and frequency subband it is in. The diagram of the wavelet decomposition principle is shown in Figure 5, where L represents low-frequency information and H represents high-frequency information. Each time the image is decomposed, the image is decomposed into four frequency bands: LL, LH, HL, and HH. The lower layer decomposition repeats the decomposition on the upper layer LL band. Wavelet transform has good performance in boundary processing of wafer defect features [21] and multi-scale edge detection [22].

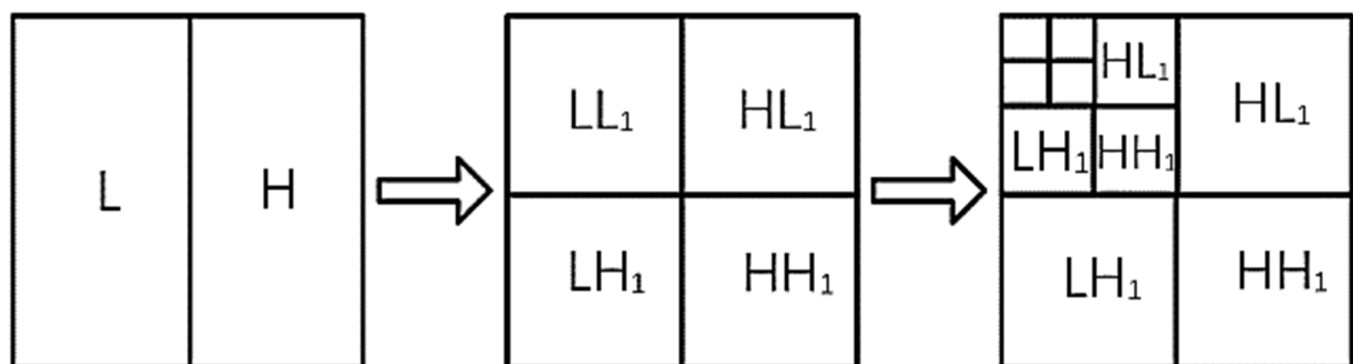

**Figure 5.** Schematic diagram of wavelet decomposition.

Yeh et al. [23] proposed a method based on two-dimensional wavelet transform (2DWT), where the ratio between scale coefficients is calculated by a modified wavelet to transform modulus (WTMS) for the localization of wafer defect pixels. By selecting the appropriate wavelet base and support length, accurate detection of wafer defects using a small amount of test data can be achieved. The image preprocessing stage takes a lot of time, which seriously affects the detection speed. Wen-Ren Yang et al. [24] proposed an online wafer microcrack detection system based on short-time discrete wavelet transform. There is no need to preprocess the wafer image. By emitting a continuous pulsed laser beam to the wafer surface, the reflected signal is collected by a space probe array, and analyzed by discrete wavelet transform to determine the reflection characteristics of microcracks. In the case of processing, it can also have a better detection effect on micro-cracks. A large number of random wafer particles exist on the surface of polycrystalline solar wafers, resulting in uneven texture in wafer sensing images. In response to this problem, Kim Y et al. [12] proposed a surface inspection method based on wavelet transform to detect solar wafer defects. To better distinguish defect edges from grain edges, the energy differences of the wavelet detail subgraphs of two consecutive decomposition levels are used as

weights to enhance the discriminative features proposed in each decomposition level. The experimental results show that this method has a good detection effect on fingerprints and stains, but the method is not effective on serious micro-crack defects with sharp edges, and cannot be applied to all defects.

### 3.2. Spatial Filtering

Spatial filtering is a mature image enhancement technique [25], which is realized by directly applying spatial convolution [26] on the gray value. The main role in image processing is image denoising, which is divided into smoothing filters [27] and sharpening filters [28], which are widely used in the field of defect detection. Figure 6 shows the denoising effect of the median filter and mean filter in the image with added noise.

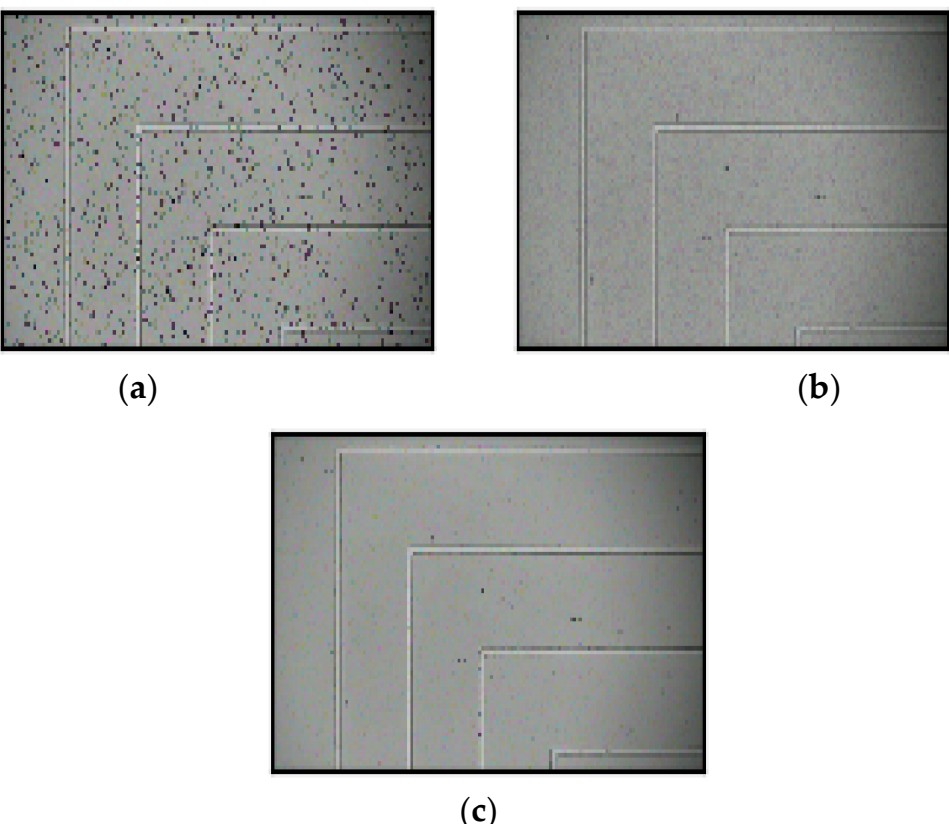

**(a)**          **(b)**

**(c)**

**Figure 6.** Effect picture of filtering and denoising: (**a**) original image, (**b**) median filter denoising, and (**c**) mean filter denoising.

Ohshige et al. [29] proposed a surface defect detection system based on spatial frequency filtering techniques. The method can effectively detect submicron defects or foreign particles on wafers. Effects of random defects in wafer manufacturing. C.H. Wang [30] proposed a wafer defect detection method based on spatial filtering, entropy-fuzzy c-means, and spectral clustering [31], which uses spatial filtering to denoise and extract defect regions, which are obtained by entropy-fuzzy c-means [29] and spectral clustering [32]. A hybrid algorithm combining mean and spectral clustering is used for defect classification. It solves the problem that traditional statistical methods cannot extract defect patterns with meaningful classification. For clustered defects in wafers, Chen SH et al. [33] developed a software tool based on median filtering and clustering methods, and the proposed algorithm effectively detected defect clusters. Often the performance of spatial filters is highly parameter-dependent, and it is often difficult to choose their values.



*3.3. Template Matching*

Template matching [36,37] detection is achieved by calculating the similarity between the template image and the image under test, to detect the difference area between the image under test and the template image.

Han H [36] et al. obtained templates from the wafer image itself mixed into the design layout scheme of the wafer fabrication process and used the mapping between the physical space and the pixel space to design a wafer image detection technology that combines existing a new method for circle template matching detection. Xifeng Liu [16] combined the SURF image registration algorithm to achieve spatial positioning matching between the test wafer and the standard wafer patterns. The feature point matching result between the test image and the standard image is shown in Figure 7. The contour extraction technique of pattern recognition is applied to wafer defect detection. Khalaj et al. [36] proposed a new technique that uses a high-resolution spectral estimation algorithm to extract wafer defect features and compare them with actual images to detect periodic 2D signals or the location of irregularities and defects in the image.

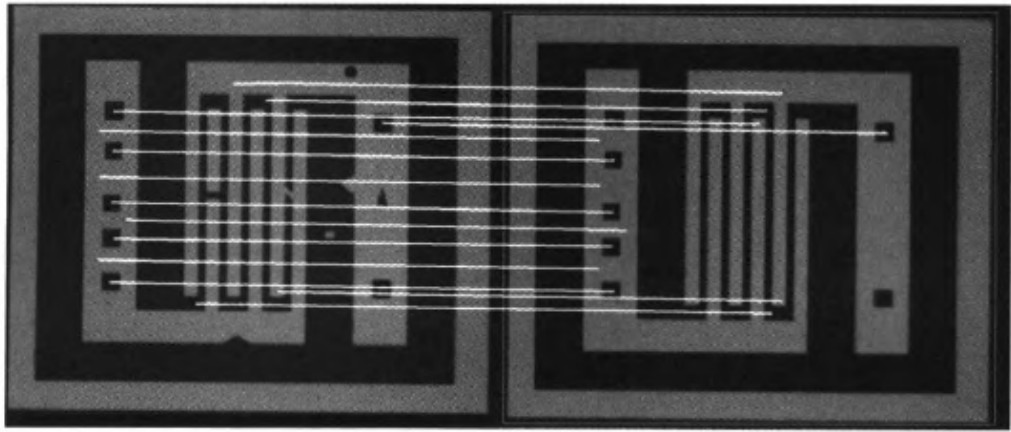

**Figure 7.** The feature point matching result between the test image and the standard image.

**4. Wafer Surface Defect Detection Based on Machine Learning**

Machine learning is mainly to abstract a specific problem into a mathematical model, solve the model through mathematical methods, solve the problem, and then evaluate the effect of the model on this problem. According to the characteristics of training data, it is divided into supervised learning, unsupervised learning, and semi-supervised learning. This paper mainly discusses the application of these three machine-learning methods in wafer surface defect detection. The machine learning model comparison is shown in Table 2.

**Table 2.** Comparison on machine learning algorithms.

| Classification | Algorithm | Innovation | Limitation |
|---|---|---|---|
| Supervised learning [38–46] | KNN | Insensitive to abnormal data and highly accurate. | High complexity and computation intensity. |
| | Decision Tree-Radon | Apply Radon to form new defect features. | Overfitting is highly proficient. |
| | SVM | SVM efficiently classifies multivariate, multi-modal, and indivisible data points. | It is not friendly to multiple samples, and the kernel function is difficult to locate. |

**Table 2.** *Cont.*

| Classification | Algorithm | Innovation | Limitation |
|---|---|---|---|
| Unsupervised learning [47–50] | Multilayer Perceptron-Clustering Algorithm | The multilayer perceptron is used to enhance the feature extraction capability. | Depends on the choice of activation function. |
| | DBSCAN | Outliers can be selectively removed based on defect pattern characteristics. | The sample density is not uniform or the sample is too large, the convergence time is long, and the clustering effect is poor. |
| | SOM | High-dimensional data can be mapped to a low dimensional space and the structure of the high-dimensional space can be maintained. | The objective function is not easy to determine. |
| Semi-supervised learning [51–53] | A semi-supervised framework for augmented labeling Semi-Supervised Increment-al Modeling Framework | A semi-supervised model is built by combining supervised ensemble learning and unsupervised SOM. Improve model performance by actively learning and labeling samples to enhance them. | Training is time-consuming and time-consuming. Performance depends on the amount of data tagged. |

### 4.1. Supervised Learning

Supervised learning is a learning model [38] created from labeled training data, and based on this model, it makes predictions about the new data samples needed. Supervised learning is currently a widely used machine learning algorithm in wafer surface defect detection and has high robustness in the field of object detection.

Yuan, T et al. [40] proposed a noise removal technique based on the k-Nearest Neighbor (KNN), which uses the k-Nearest Neighbor algorithm to separate global and local defects, providing information about all aggregated local defects of the wafer information, classify defects into clusters through similarity clustering techniques, and identify spatial patterns of defect clusters using a parametric model of clustered defects. Piao M et al. [41] proposed a decision tree-based wafer defect pattern recognition method. The radon transformation is used to extract the defect pattern features, the correlation analysis method is used to measure the correlation between the features, and the defect features are divided into feature subsets, and each feature subset builds a decision tree according to the C4.5 mechanism. The decision tree confidences are summed and the category with the highest overall confidence is selected. The decision tree shows better performance in specific categories of wafer defect detection, but the maximum, minimum, average, and standard deviation of projections are not enough to represent all the spatial information of wafer defects, so the edge defect detection performance is poor.

Support Vector Machine (SVM) in supervised learning is also a mature application of defect detection. When the samples are unbalanced, the k-nearest neighbor algorithm has a poor classification effect and a large amount of computation. Decision trees have similar problems and are prone to overfitting. The support vector machine still has good performance in the classification of small samples and high-dimensional features, and the computational complexity of the support vector machine does not depend on the dimension of the input space, and the multi-class support vector machines are robust to overfitting problems, so it is often used as a classifier [42–44]. R. Baly et al. [45] used a Support Vector Machine (SVM) classifier to classify 1150 wafer images into two categories, high-yield and low-yield, and then demonstrated through comparative experiments that relative to decision trees, k-Nearest Neighbor (KNN), Partial Least Squares Regression (PLS regression) and Generalized Regression Neural Network (GRNN), the nonlinear support vector machine model is better than the above four wafer classification methods. The multi-class SVM has better classification accuracy in the classification of wafer defect patterns. L. Xie et al. [46] proposed a wafer defect pattern detection scheme based on a support vector machine algorithm. The linear kernel, Gaussian kernel, and polynomial kernel are

used for selective testing, and the kernel with the smallest test error is selected through cross-validation for the next step of support vector machine training. The support vector machine method can deal with the problem of false positives caused by image translation or rotation. Compared with neural networks, SVM does not require a large number of training samples, so it does not need to spend a lot of time training data samples for classification. More robust performance for composite or diverse datasets.

### 4.2. Unsupervised Learning

In supervised learning, researchers need to classify defect sample types in advance as prior knowledge for training. In actual industrial production, there are a large number of unknown defects and the defect characteristics are ambiguous, and it is difficult for researchers to judge and classify through experience. In the early stages of process development, sample annotation is also limited. Aiming at these problems, unsupervised learning [47] has opened up new solutions, which do not need a lot of manpower for labeling data samples and performing clustering according to the feature relationship between samples. Unsupervised learning also has advantages when new defect patterns are added. In recent years, unsupervised learning has become one of the important research directions of industrial defect detection.

The defect patterns on the wafer pattern are unevenly classified and the features are irregular, and the unsupervised clustering algorithm has strong robustness against this situation and is widely used to detect complex wafer defect patterns. Due to the difficulty in detection as a result of cluster defects, such as scratches, stains, or local failure modes, C. J. Huang [48] proposed a new method to solve this problem. An automatic wafer defect clustering algorithm (k-means clustering) using self-supervised multilayer perceptrons to detect defects and label all defective chips was proposed. Jin C H et al. [49] proposed a wafer pattern detection and classification framework based on Density-Based Spatial Clustering of Applications with Noise (DBSCAN), which selectively removed outliers according to defect pattern features, and then the extracted defect features can complete the detection of abnormal points and defect patterns at the same time. Yuan, T et al. [40] proposed a multi-step wafer analysis method, which provides clustering results with different precisions based on a similarity clustering technique to identify six mixed-type defect patterns according to the spatial location of local defect patterns. Using location information to distinguish defect clusters has certain limitations, and when multiple clusters are close to or overlap with each other, the classification effect will be affected.

Di Palma, F et al. [50] employed unsupervised Self-Organizing Map (SOM) and Adaptive Resonance Theory (ART1) as wafer classifiers on nine different classes of wafers tested on a simulated dataset. Both SOM and ART1 rely on the competition between neurons to gradually optimize the network for unsupervised classification. Since ART1 is pushed to the reference vector by "AND" logic, when processing a large number of data sets, the number of calculations increases, and the real number of defect categories cannot be obtained. Adjusting the network identification threshold does not make any improvement. The SOM algorithm can map high-dimensional input data to a low-dimensional space while maintaining the topological structure of the input data in the high-dimensional space. First, the category and number of neurons are determined, and other parameters are determined through several comparison experiments. After determining the parameters, after several learning cycles, the data reaches an asymptotic value and performs well on both simulated and real datasets.

### 4.3. Semi-Supervised Learning

Semi-supervised learning is a machine learning method that combines supervised learning and unsupervised learning [51]. Semi-supervised learning can use a small amount of labeled data and a large amount of unlabeled data to solve problems. The ensemble-based semi-supervised learning process is shown in Figure 8. The cost consumption and

mislabeling of fully labeled samples are avoided. Semi-supervised learning has become a hot topic of research in recent years.

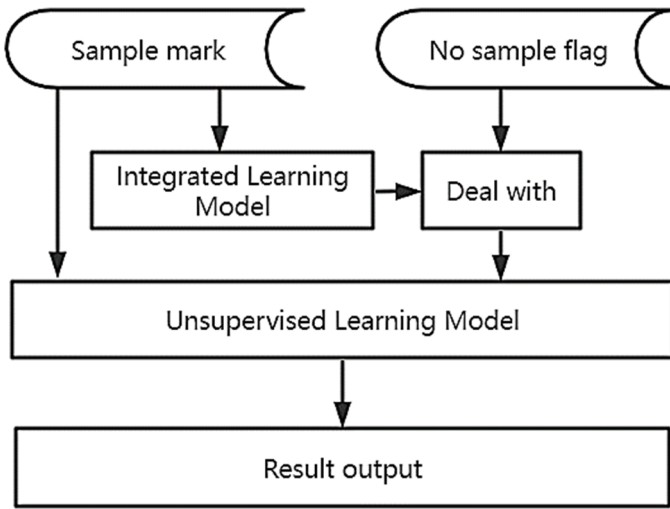

**Figure 8.** Ensemble-based semi-supervised learning.

Supervised learning usually achieves good recognition results, but relies on the accuracy of sample labeling. The wafer data samples may have the following problems. The first is that the wafer sample data needs to be manually marked by professionals. The manual marking process is subjective, and some mixed defect modes may be incorrectly marked. The second is that some defect modes have insufficient samples. The third is that some defect patterns are not marked in the first place. As a result, unsupervised learning methods cannot exert their performance. Aiming at this problem, Katherine Shu-Min Li [52] et al. proposed an ensemble-based semi-supervised framework to realize the automatic classification of defect patterns. First, a supervised ensemble learning model was trained on labeled data, and then the unlabeled data were trained by this model. Finally, the unsupervised learning algorithm is used to process the samples that cannot be correctly classified to achieve an enhanced labeling effect and improve the accuracy of wafer defect pattern classification. Yuting Kong and Dong Ni [53] proposed a semi-supervised incremental modeling framework for wafer map analysis. A semi-supervised incremental model improved by a ladder network and SVAE model is used to classify wafer maps, and then the model performance is improved by active learning and pseudo-labeling. Experiments show that it has a better performance than the CNN model.

## 5. Wafer Surface Defect Detection Based on Deep Learning

In recent years, with the development of deep learning algorithms, the improvement of GPU computing power, and the emergence of the convolutional neural network [54], the field of computer vision has been qualitatively developed, and it has also been widely used in the field of surface defect detection. Before deep learning, relevant personnel were required to have extensive knowledge of feature mapping and feature description to be able to draw features manually. Deep learning enables multi-layer neural networks to automatically extract and learn target features through abstraction layers, and detect target objects from images.

Cheng KCC et al. [55] used machine learning algorithms and deep learning algorithms for wafer defect detection, respectively. They used Logistic Regression [56], Support Vector Machine (SVM) [57], Adaptive Boosting Decision Tree (ADBT) [58], and deep neural networks to detect wafer defects. Experiments have proved that the average accuracy of the deep neural network is better than the above machine learning algorithms, and the wafer detection algorithm based on deep learning has a better performance. According to different application scenarios and task requirements, the deep learning model is di-

vided into a classification network, detection network, and segmentation network. This section discusses the innovations and compares the performance of each deep learning network model.

### 5.1. Classification Network

Classification networks are one of the older deep learning algorithms. The classification network extracts the characteristic information of the target object in the input image through a series of operations, such as convolution [54] and pooling [58], and then passes through the fully connected layer [59] to classify according to the preset label category. The network model is shown in Figure 9. In recent years, many classification networks for specific problems have emerged. In the field of wafer defect detection, focusing on defect features and enhancing feature extraction capabilities has promoted the development of wafer detection.

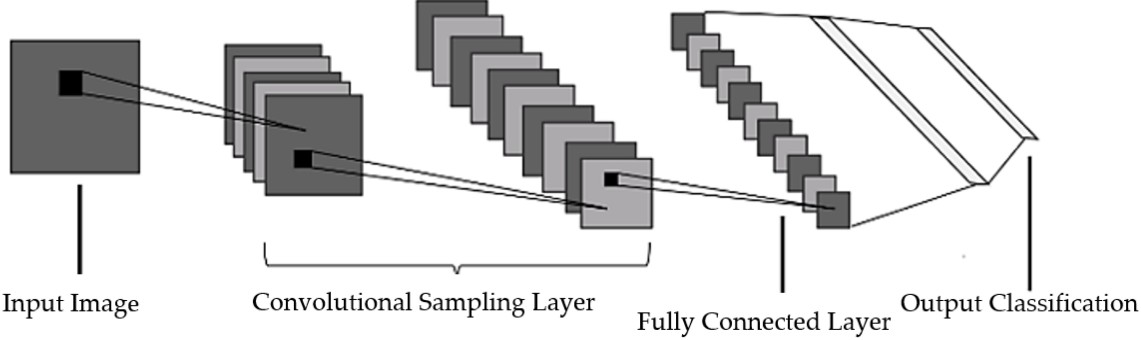

**Figure 9.** Classification network model structure diagram.

In the wafer manufacturing process, several different types of defects are coupled in a wafer, called mixed defects. These types of defects are complex, changeable, and highly random, and have become a key challenge for semiconductor companies. In response to this problem, Wang J et al. [60] proposed a hybrid DPR (MDPR) deformable convolutional network (DC-Net) for wafer defect classification. They designed a multi-label output of deformable convolution and a one-hot encoding mechanism layer, focusing the sampling area on the defect feature area, effectively extracting defect features, classifying mixed defects, outputting single defects, and improving the classification accuracy of mixed defects. Kyeong and Kim [61] designed a separate classification model for each type of defect in the wafer image of the mixed defect mode and detected the defect mode of the wafer through the combined classifier network. The authors tested the wafer map database of six different modes with MPL, SVM, and CNN combined classifiers, and only the algorithm proposed by the authors was classified correctly. Takeshi Nakazawa and Deepak V. Kulkarni [62] used CNN for wafer defect pattern classification. They trained and validated their CNN model using synthetically generated wafer images. Additionally, a method of using simulated generated data was proposed to solve the problem of unbalanced data of real defect categories in manufacturing and achieve reasonable classification accuracy. This effectively solves the problem of difficult wafer data collection and few available samples. The classification network model comparison is shown in Table 3.

**Table 3.** Classification network model comparison.

| Algorithm | Innovation | Acc |
|---|---|---|
| DC-Net [60] | The sampling area is focused on the defect feature area, which is very robust to mixed defects. | 93.2% |
| CNN-Based Combined Classifier [61] | Separately design classifiers for each defect, strong adaptability to new defect modes. | 97.4% |

**Table 3.** *Cont.*

| Algorithm | Innovation | Acc |
|---|---|---|
| Classification Retrieval Method Based on CNN [62] | Simulated datasets can be generated to account for data imbalances. | 98.2% |

*5.2. Object Detection Network*

The target detection network can not only classify the target object but also identify its location. The target detection network is mainly divided into two types. The first type is the two-stage network [63,64], as shown in Figure 10. The candidate boxes are generated based on the region proposal network, and then the candidate boxes are classified and regressed. The second category is single-stage networks [65–67], as shown in Figure 11, that is, end-to-end object detection, which directly generates classification and regression information of target objects without generating candidate boxes. Relatively speaking, the two-stage network has a higher detection accuracy, and the single-stage network has a faster detection speed. The comparison of the detection network models is shown in Table 4.

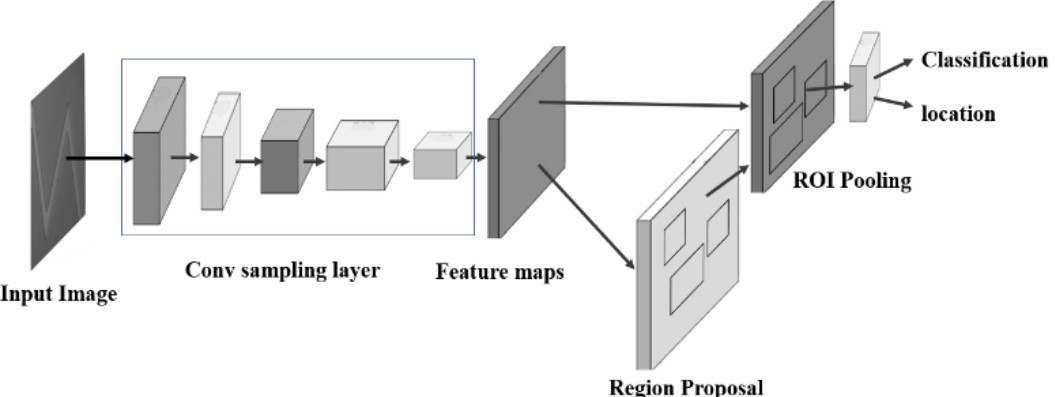

**Figure 10.** Schematic diagram of the structure of the two-stage detection network model.

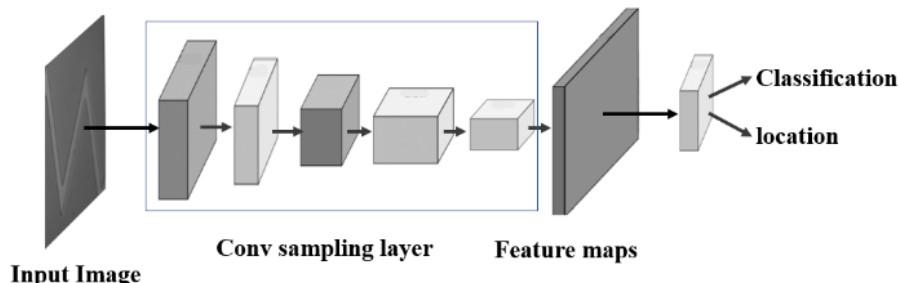

**Figure 11.** Schematic diagram of the structure of the one-stage detection network model.

**Table 4.** Detection network model comparison.

| Algorithm | Innovation | Acc | Ap |
|---|---|---|---|
| PCACAE [68] | Automatic coding of concatenated roll types based on two-dimensional principal component analysis. | 97.27% | \ |
| YOLOv3-GAN [33] | GAN enhances the diversity of defect patterns and improves the versatility of YOLOv3. | \ | 88.72% |
| YOLOv4 [69] | Updated backbone network, enhanced with CutMix and Mosaic data. | 94.0% | 75.8% |

Yu J et al. [68] proposed a deep neural network PCACAE based on a convolutional autoencoder based on two-dimensional principal component analysis and designed a new convolution kernel for extracting wafer defect features. The product autoencoder is cascaded to further improve the performance of feature extraction. Aiming at the problems of difficult collection of wafer data and few public datasets, Ssu-Han Chen et al. [33] used a combination of the generative adversarial network and target detection algorithm YOLOv3 for the first time to detect wafer defects in small samples. The diversity of defects is enhanced by GAN, which improves the generalization ability of YOLOv3. Prashant P. SHINDE et al. [69] proposed the use of advanced YOLOv4 to detect and locate wafer defects. Compared with YOLOv3, the backbone extraction network was improved from Darknet-19 to Darknet-53, and the mish activation function was used to make the network robust. The stickiness is enhanced, the detection ability is greatly improved, and the detection and positioning performance of complex wafer defect modes is more efficient.

*5.3. Segment Network*

Segmentation networks perform pixel-level segmentation of regions of interest in input images [70]. Most of the segmentation network is based on the structure of the encoder and decoder, as shown in Figure 12 is a schematic diagram of the segmentation network model structure. Through the encoder and decoder, the ability to extract the features of the target object is improved, and the analysis and understanding of the image by the subsequent classification network are strengthened. It has a good prospect to be applied to wafer surface defect detection.

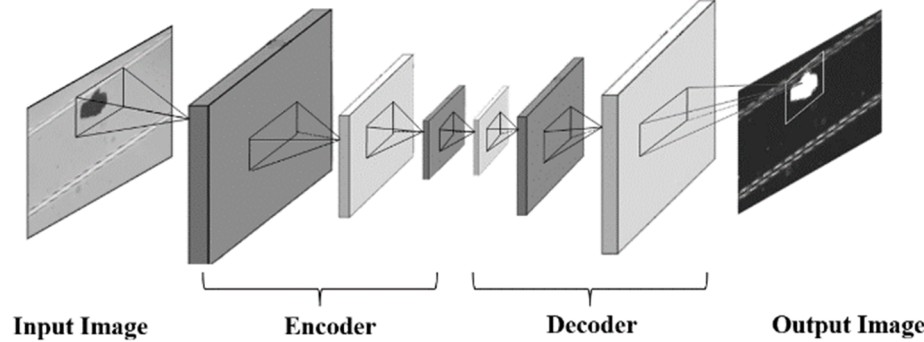

**Figure 12.** Schematic diagram of segmentation network model structure.

Takeshi Nakazawa et al. [70] proposed a deep convolutional encoder–decoder neural network structure for anomaly detection and segmentation of wafer defect patterns. The author designed three encoder–decoder wafer defect pattern segmentation networks based on FCN, U-Net, and SegNet to segment the wafer, local defect model. Global random defects in the wafer often lead to noise in the extracted features. After segmentation, the influence of global defects on local defects is ignored, and more information about defect clusters is helpful for further analysis of the reasons. Aiming at the problem of unbalanced wafer defect pixel categories and insufficient samples, Han Hui et al. [36] designed an improved segmentation system based on a U-net network. Based on the original UNet network, the RPN network is added to obtain defect area suggestions, and then input to the Unit network for segmentation. The designed two-stage network has an accurate segmentation effect on wafer defects. Subhrajit Nag [71] et al. proposed a new network structure WaferSegClassNet with a decoder–encoder architecture. The encoder extracts better multi-scale local details through a series of convolutional blocks, and the decoder is used to perform classification and generation. Segmentation mask, which is the first wafer defect detection model that can classify and segment at the same time, has a good segmentation and classification effect for mixed wafer defects. The segmentation network model comparison is shown in Table 5.

**Table 5.** Segmentation network model comparison.

| Algorithm | Innovation | Acc |
|---|---|---|
| FCN [62] | Replacing fully connected layers with convolutional layers to output 2D heatmaps. | 97.8% |
| SegNet [62] | Combining encoder–decoder and pixel-level classification layers. | 99.0% |
| U-net [36] | Copy and crop the feature maps in each encoder layer to the corresponding decoder layer. | 98.9% |
| WaferSegClassNet [70] | Simultaneous classification and Segmentation using shared encoders. | 98.2% |

## 6. Conclusions and Outlook

With the continuous development of electronic information technology and the continuous improvement of lithography technology, wafer surface defect detection occupies an important position in the semiconductor industry, and has attracted increasing attention from scholars in this field. This paper analyzes and summarizes the research on image signal processing, machine learning, and deep learning related to wafer surface defect detection. In the early days, image signal processing methods were mainly used, among which wavelet transform methods and spatial filtering methods were widely used. Machine learning is very robust in wafer defect detection. Algorithms such as k-Nearest Neighbor (KNN), Decision Tree (Decision Tree), and Support Vector Machine (SVM) are widely used in this field and have achieved good results. Deep learning has brought vitality to the field of wafer inspection with its powerful feature extraction capabilities. The latest manufacturing technology for integrated circuits has advanced to 4 nm, and predictions suggest it will continue to develop towards even smaller scales. However, as these trends emerge, the complexity of surface defects on wafers will also increase, posing more stringent challenges to the reliability and robustness of models. Therefore, the analysis and processing of these defects become increasingly important to ensure high-quality manufacturing of integrated circuits. Although some achievements have been made in the field of wafer surface defect analysis, there are still many problems and challenges.

1. There are few public datasets of wafer defects. Due to the high cost of wafer production and labeling, there are very few high-quality public datasets, and the few datasets are not enough to support training. It is possible to consider creating a synthetic wafer defect database and performing data augmentation on the existing dataset to provide more accurate and comprehensive data samples for neural networks. Due to the versatility of defect types in gradient features, such problems can be addressed using transfer learning, mainly to solve problems such as negative transfer and model inappropriateness in transfer learning [72]. A flexible and efficient migration model does not currently exist. Using transfer learning to solve the problem of a few samples in wafer surface defect detection is a difficult topic for future research.

2. During the wafer fabrication process, new defects are continuously generated, and the number and types of defect samples are continuously accumulated. Using incremental learning [73] can improve the recognition accuracy of the network model for new defects and the ability to maintain the classification of old defects. It can also be used as a research direction for the expanded sample method.

3. With the rapid development of technological progress, the chip feature size is becoming smaller and more complex, resulting in multiple defect types in a wafer, and the defects are folded with each other, resulting in non-uniform and inconspicuous defect features. increase the difficulty of detection. Multi-step, multi-method hybrid models have become the mainstream method for detecting hybrid defects. How to optimize the performance of the deep network model and maintain a high detection efficiency is a problem that needs to be further solved.

4. During the wafer fabrication process, wafer patterns for different purposes will produce different defects. Currently, a network model trained on a single data set is



not sufficient to identify defects in all wafers for different purposes. How to design a universal network model to detect all defects, thereby avoiding the waste of resources caused by designing a training model separately for all wafer defect data sets, is a direction worth thinking about in the future.

5. The majority of defect detection models are offline models, which are unable to meet the real-time requirements of industrial production. To address this issue, an autonomous learning model system needs to be established, which enables the model to rapidly learn and adapt to new production environments, thereby achieving more efficient and accurate defect detection.

**Author Contributions:** Conceptualization, J.M.; writing—review and editing, T.Z.; investigation, C.Y.; resources, Y.C.; writing—review and editing, H.T.; supervision, L.X.; project administration, X.L. All authors have read and agreed to the published version of the manuscript.

**Funding:** This research was funded by National Key Research and Development Program Key Special Project (2020YFB1712401-1), ZhengZhou Collaborative Innovation Major Project (20XTZX06013), China Postdoctoral Science Foundation (2021M692926), HeNan Science and Technology Research (222102310647) and Strategic Research and Consulting Project of Chinese Academy of Engineering (2022HENYB03).

**Data Availability Statement:** Not applicable.

**Conflicts of Interest:** The authors declare no conflict of interest.

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
