# Peer review of "Review of Wafer Surface Defect Detection Methods"

_electronics, doi:10.3390/electronics12081787_

Round 1

Reviewer 1 Report

The submitted manuscript presents a review of techniques for detecting surface defects in wafers. However, several articles in the state of the art have not been considered, such as:

Wang, X., Jia, X., Jiang, C., & Jiang, S. (2022). A wafer surface defect detection method built on generic object detection network. Digital Signal Processing, 130, 103718.

Zhu, J., Liu, J., Xu, T., Yuan, S., Zhang, Z., Jiang, H., ... & Liu, S. (2022). Optical wafer defect inspection at the 10 nm technology node and beyond. International Journal of Extreme Manufacturing.

Shi, Z., Sang, M., Huang, Y., Xing, L., & Liu, T. (2022). Defect detection of MEMS based on data augmentation, WGAN-DIV-DC, and a YOLOv5 model. Sensors, 22(23), 9400.

Kim, T., & Behdinan, K. (2022). Advances in machine learning and deep learning applications towards wafer map defect recognition and classification: a review. Journal of Intelligent Manufacturing, 1-33.

Guan, J., Li, J., Yang, X., Chen, X., & Xi, J. (2022). Defect detection method for specular surfaces based on deflectometry and deep learning. Optical Engineering, 61(6), 061407-061407.

Chen, P. C., Miao, W. C., Ahmed, T., Pan, Y. Y., Lin, C. L., Chen, S. C., ... & Lien, D. H. (2022). Defect inspection techniques in SiC. Nanoscale Research Letters, 17(1), 1-17.

It is suggested to consider some of these references within the manuscript and their inclusion in the introduction section.

Figure 1 should be enlarged to better visualize the defects because Figure 1a) also shows the defect of contamination by particles.

Several paragraph ideas at different parts of the manuscript are incomplete, like:

- lines 100 to 113

-"Circle chart dataset" at line 116.

-"Articles cited in this article, This wafer map dataset is mostly used" at lines 118 and 119.

It is suggested to correct the above incomplete sentences.

The word "Texture" at line 140 should be changed to "texture".

In section 3.1, related to wavelet analysis, the authors forgot to mention the type of wavelet used for this purpose, like morlet, paul, shannon, haar, and so on. Also, the strategy used in this type of analysis is not mentioned, like in the next article:

Santoyo, J., Ortega, J. C. P., Mejía, L. F., & Santoyo, A. (2007, November). PCB inspection using image processing and wavelet transform. In MICAI (pp. 634-639).

Throughout the manuscript, the idea of generating or creating synthetic databases was not mentioned, which would help to train techniques based on artificial neural networks. Another proposal is the use of data augmentation techniques when the datasets are limited.

It is suggested to include a paragraph where new techniques or strategies for detecting defects in wafers can be identified.

Reviewer 2 Report

In this review, the authors studied the possible defects in the Wafer Surface and also studied the different methods to detect these defects. The manuscript is suggested to be accepted after the following issues are addressed.   

1)      There are many spelling, grammatical, and typo errors, the authors should double check and revise them thoroughly. For example “Wafers are silicon wafers used in the manufacture of semiconductor chips.” Etc.

2)      Recent developments on this topic with critical discussion on the future perspectives should be provided. In the present version, the review is written by considering the literature survey only.

3)      What's the novelty of this review? The authors should describe in abstract and conclusion in details with future perspective.

4)      The references are not up-to-date. Most of the references are from last 3 years. The authors should add some latest work

5)      It better to increase the number of figures. The authors should add figures in each section.

more comments:

1) On page 2, line 62, the authors mentioned different types of defects in Figure 2. But there is no information about the defect's origin, nature and solutions. The authors should describe this in detail.

2) Figure 1 and 2 are not proper for review articles. The authors should revise the figures.

3) On page 3, the authors mentioned that “Defects caused by specific reasons appear repeatedly on the wafer……… the authors should explain these reasons in detail

4)      In the literature, there are three methods for detecting surface defects: texture, surface and shape. So, the authors should use these features to explain these defects

5) They should add a section such as; the key problems and mention the problems such as real-time problems; minor target detection problems

Reviewer 3 Report

The paper is interesting and deals with an important topic in the Wafer surface defect detection domain. According to the characteristics of the algorithm, the relevant literature is summarized and sorted out, and the problems and challenges faced in the field of wafer defect detection and future development have been prospected. The paper needs revision before it is ready for publication. Please find below some comments to that will improve the paper.

1. The motivation of the problem statement is not clear. The authors should provide a clear motivation of why the contribution is needed.

2. Introduction is very concise, does not provide sufficient background and does not  include sufficient references. The problem should be stated more clearly and in more detail, with more supporting references.

3. The novelty of the paper is not clear, recommend author to revise the write up and revisit the details.

Reviewer 4 Report

This study proposes a review of wafer surface defect detection methods available. Its main contribution consists in analyzing and dividing the methods based on: image signal processing, machine learning and deep learning.

The document is sometimes hard to read and follow.

The English needs major revision and spell checking.

The document is well supported with references although the majority are old. As authors say, since it is a hot topic, it would be expected more recent references.

The proposed work main weakness is the lack of a clear reasoning along the document and the lack of a deeper analysis of the presented methods. Since the document is a review, authors should also have included an in-depth comparison and discussion of the presented methods.

Below, authors can find some examples of the many typos found in the document: 

In line 8 did you mean “literature review”?!

In line 89 did you mean Figure 3?!

In line 90 did you mean Figure 3(b) ?!

In line 164 please correct “… detect and detect…”

In line 295 and 296 please correct “…After 295 determining the parameters, after ten After several learning cycles…”

In line 316 it is not clear when authors say that “…the unlabeled data were trained by this model”. Please correct.

Table 3 is introduced before being referenced in the text. Please correct.

Tables and Figures must be introduced in the document only after being referenced in the text.

In Table 4 there are some missing percentages. How do authors compare methods with missing Acc and Ap?!

Round 2

Reviewer 2 Report

Accept in present form.

Reviewer 3 Report

The revised manuscript can be accepted for publication in Electronics, however, it needs to be carefully edited before publication.  I accept the job and congratulations. Best regards

Reviewer 4 Report

Since the authors addressed the main issues pointed out in the previous review i advise that the manuscript should be accepted for publication.